

# Open source approaches to establishing *Roseobacter* clade bacteria as synthetic biology chassis for biogeoengineering

Yanika Borg[1,2,*], Aurelija Marija Grigonyte[3,*], Philipp Boeing[4], Bethan Wolfenden[4], Patrick Smith[5], William Beaufoy[5], Simon Rose[5], Tonderai Ratisai[5], Alexey Zaikin[2,6] and Darren N. Nesbeth[1]

[1] Department of Biochemical Engineering, University College London, United Kingdom
[2] Department of Mathematics, University College London, London, United Kingdom
[3] Synthetic Biology Centre for Doctoral Training, University of Warwick, Coventry, United Kingdom
[4] Bento Bioworks, UCL Advances, London, United Kingdom
[5] London BioHackspace, London, United Kingdom
[6] Institute for Women's Health, University College London, London, United Kingdom
[*] These authors contributed equally to this work.

Corresponding author
Darren N. Nesbeth,
d.nesbeth@ucl.ac.uk

## ABSTRACT

**Aim.** The nascent field of bio-geoengineering stands to benefit from synthetic biologists' efforts to standardise, and in so doing democratise, biomolecular research methods. *Roseobacter* clade bacteria comprise 15–20% of oceanic bacterio-plankton communities, making them a prime candidate for establishment of synthetic biology chassis for bio-geoengineering activities such as bioremediation of oceanic waste plastic. Developments such as the increasing affordability of DNA synthesis and laboratory automation continue to foster the establishment of a global 'do-it-yourself' research community alongside the more traditional arenas of academe and industry. As a collaborative group of citizen, student and professional scientists we sought to test the following hypotheses: (i) that an incubator capable of cultivating bacterial cells can be constructed entirely from non-laboratory items, (ii) that marine bacteria from the *Roseobacter* clade can be established as a genetically tractable synthetic biology chassis using plasmids conforming to the BioBrick[TM] standard and finally, (iii) that identifying and subcloning genes from a *Roseobacter* clade species can readily by achieved by citizen scientists using open source cloning and bioinformatic tools.

**Method.** We cultivated three *Roseobacter* species, *Roseobacter denitrificans*, *Oceanobulbus indolifex* and *Dinoroseobacter shibae*. For each species we measured chloramphenicol sensitivity, viability over 11 weeks of glycerol-based cryopreservation and tested the effectiveness of a series of electroporation and heat shock protocols for transformation using a variety of plasmid types. We also attempted construction of an incubator-shaker device using only publicly available components. Finally, a subgroup comprising citizen scientists designed and attempted a procedure for isolating the cold resistance *anf1* gene from *Oceanobulbus indolifex* cells and subcloning it into a BioBrick[TM] formatted plasmid.

**Results.** All species were stable over 11 weeks of glycerol cryopreservation, sensitive to 17 µg/mL chloramphenicol and resistant to transformation using the conditions and plasmids tested. An incubator-shaker device, 'UCLHack-12' was assembled and used

to cultivate sufficient quantity of *Oceanobulbus indolifex* cells to enable isolation of the *anf1* gene and its subcloning into a plasmid to generate the BioBrick[TM] BBa_K729016. **Conclusion.** The process of 'de-skilling' biomolecular techniques, particularly for relatively under-investigated organisms, is still on-going. However, our successful cell growth and DNA manipulation experiments serve to indicate the types of capabilities that are now available to citizen scientists. Science democratised in this way can make a positive contribution to the debate around the use of bio-geoengineering to address oceanic pollution or climate change.

# INTRODUCTION

The last decade has seen increased discussion as to whether global phenomena that result from human activity, such as climate change (*Rayner et al.*, *2013*) and oceanic pollution (*Hale & Dilling*, *2011*), can and should be met with geoengineering (*Stilgoe*, *2015*; *IMBECS*, *2014*) and bio-geoengineering (*Singarayer & Davies-Barnard*, *2012*) solutions. Synthetic biology has begun to feature in this field due to developments such as the proposed used of gene drives (*Jin et al.*, *2013*) to control insect populations in the wild. Conventional bio-geoengineering proposals involve the re-seeding of naturally occurring organisms, such as certain barley varieties, in non-native geographical locations to increase global solar reflectivity (*Ridgwell et al.*, *2009*).

A major challenge for synthetic biology approaches to bio-geoengineering is the establishment of organisms, or 'chassis', that are viable in natural habitats. Natural environments tend to be physically and chemically harsh and possess only scarce nutrient sources. This contrasts with the laboratory environment, which is constantly monitored, maintained and optimised to achieve maximal growth of laboratory-adapted organisms such as *E. coli* K-12 (*Bachmann*, *1972*), *P. pastoris* GS115 (*De Schutter et al.*, *2009*) and Chinese hamster ovary cells (*Xu et al.*, *2011*).

In this study we seek to exploit properties of a clade of marine bacteria, *Roseobacter* (*Brinkhoff, Giebel & Simon*, *2008*), as a chassis for marine applications of synthetic biology and a source of genetic material that could be used to confer upon more conventional chassis, such as *E. coli*, the ability to grow in a marine environment. We anticipate that establishing standard tools to engineer marine bacteria could underpin the future deployment of a designed organism in the world's oceans capable of sensing and degrading the waste plastics observed to accumulate in oceanic gyres (*Eriksen et al.*, *2014*). We also anticipate that any such bio-geoengineering steps would only be taken with broad societal consent, of the type described by *Stilgoe* (*2015*) and others (*Rayner et al.*, *2013*). As an interdisciplinary team of professional scientists, student scientists and citizen scientists, we embarked on this study as a means of exploring the logistical, scientific, didactic and ethical challenges and opportunities presented by scientific research practiced by members of the public in a non-conventional research setting.

*Roseobacter* is one of the nine major clades of marine bacteria (*Buchan, González & Moran*, *2005*) that provide the vast bacterial diversity present in the world's oceans. *Roseobacter* can represent up to a fifth of the total species present in bacterio-plankton communities at certain oceanic depths and periods within a given year (*González, Kiene & Moran*, *1999*; *Wagner-Döbler & Biebl*, *2006*). Due to their extreme versatility, *Roseobacter* clade bacteria can survive in aerobic and anaerobic environments, interact with eukaryotic cells via symbiosis (*Buchan, González & Moran*, *2005*), utilise quorum-sensing mechanisms (*Zan et al.*, *2014*), facilitate the oxidation of carbon monoxide to carbon dioxide (*Brinkhoff, Giebel & Simon*, *2008*) and also produce dimethylsulfide, a key component of the global sulphur cycle (*Hahnke et al.*, *2013*).

The genomes of over 40 *Roseobacter* strains have been sequenced (*Petersen et al.*, *2013*) but only one group has demonstrated transformation of *Roseobacter* species with recombinant plasmids (*Piekarski et al.*, *2009*). Establishing a *Roseobacter* strain that is sufficiently genetically tractable to be used in 'de-skilled', robust and reliable modification protocols could enable the application of designed organisms to address pressing challenges such as climate change (*Ridgwell et al.*, *2009*) and plastic pollution (*Dash et al.*, *2013*). 50–80% of sea-debris on beaches, the seabed and floating in the ocean has been estimated to consist of micro-plastics (*Barnes et al.*, *2009*; *Hidalgo-Ruz & Thiel*, *2013*). Marine bacteria controlled by synthetic gene networks or genomes have the potential to be used as plastic remediation systems that utilise laccases, enzymes capable of degrading polyethylene into non-hazardous polymers (*Santo, Weitsman & Sivan*, *2013*).

A significant step in establishing a bacterial species as a synthetic biology chassis is to establish whether plasmids compliant with the BioBrick$^{TM}$ format can be used for transformation. The BioBrick$^{TM}$ plasmid format is shared by all parts (plasmids) available from the open Registry of Standard Biological Parts, which is maintained by the International Genetically Engineered Machines (iGEM) Foundation (Boston, USA). This registry is a large plasmid library maintained by the staff of the iGEM annual synthetic biology competition (*Müller & Arndt*, *2012*) and curated in part by users. Every plasmid in the BioBrick$^{TM}$ format is compatible with every other BioBrick$^{TM}$ and so demonstrating the use of one BioBrick$^{TM}$ plasmid in a *Roseobacter* species immediately enables the use of several thousand compatible plasmid-based tools to build synthetic genes and gene networks. A given DNA sequence can be classified as a BioBrick$^{TM}$ part if it is flanked upstream by a defined sequence motif which encodes, in order, EcoRI, NotI and XbaI restriction sites and flanked downstream by a sequence encoding, in order, unique SpeI, NotI and PstI sites (*Canton, Labno & Endy*, *2008*; *Shetty et al.*, *2011*).

The BioBrick$^{TM}$ format enables recursive rounds of DNA ligation in which the enzymes and procedures used do not change, regardless of the identity of the underlying fragments being assembled. This approach enables an interchangeable 'plug-and-play' strategy for mixing and matching genetic components within a gene or genes within a pathway. This typically results in a more predictable, economically viable, and time efficient practice than conventional *ad hoc* recombinant DNA strategies (*Tabor*, *2012*). In this study we were keen to investigate whether a simple and affordable plasmid transformation procedure

**Table 1 Plasmids used in transformation study.** All plasmids were sourced directly from people or organisation listed except pHD1313, which was a kind gift from Prof. Christine Clayton (Universität Heidelberg).

| | Name | *Ori* | Selection | Source/Reference |
|---|---|---|---|---|
| 1 | pSB3C5 | p15A | Chloramphenicol | BioBrick™ Registry |
| 2 | pA0815 | pBR322 | Ampicillin | Thermo Fisher Cat. No. V18020 |
| 3 | pHD1313 | pUC | Ampicillin | *Alibu et al.* (*2005*) |
| 4 | pRP^GFP SIR2rp3 | pUC | Ampicillin | *Borg* (*2015*) |
| 5 | pUBeK | pUC | Ampicillin | *Borg* (*2015*) |

could be established in the *Roseobacter* strains, *Roseobacter denitrificans* (*R. denitrificans*), *Oceanobulbus indolifex* (*O. indolifex*) and *Dinoroseobacter shibae* (*D. shibae*). Tolerance to cold is also a potentially useful phenotype to port from *Roseobacter* to *E. coli*. Toward this end we also attempted to isolate the gene OIHEL45_03590, encoding Antifreeze protein type I (referred to here as *anf1*) from *O. indolifex*, and subclone it into a BioBrick™ plasmid.

In addition to streamlining and enhancing recombinant DNA procedures in conventional research settings, the BioBrick™ standard for plasmid design and assembly can also help provide the reproducibility and robustness that enables a methodology to be accessible to members of the public participating in citizen science projects (*Wolyniak et al.*, *2010*). Citizen science has led to the public's involvement in a variety of ecology (*Shirk et al.*, *2012*), conservation (*Hochachka et al.*, *2012*), biology (*Jordan et al.*, *2011*) and genetics (*Kawrykow et al.*, *2012*) projects. Known alternatively as 'biohackers', 'citizen scientists', 'garage scientists' and 'DIY biologists' (*Ledford*, *2010*), growing numbers of people are now taking advantage of open source software and hardware in biological research. Commonly used open source devices used for 'biohacking' include Arduino prototyping platforms, 3D-printers, spectrophotometers (iGEM, Boston, MA, USA), thermal cyclers (Open PCR; Chai Biotechnologies Inc., Santa Clara, CA) for end-point polymerase chain reactions (epPCR) and thermal cyclers with live fluorescence detection (Chai Biotechnologies Inc., Santa Clara, CA), for quantitative PCR (qPCR). The fact that these tools are open source means that well-equipped laboratories can now be found outside of both academia and industry, in community-based spaces (*Alper*, *2009*; *Pearce*, *2012*).

Alternative funding models are increasingly being used to support the running of projects and community laboratories through 'crowdfunding' via online companies that act as intermediaries to enable private individuals to invest in projects or propose ventures (*Belleflamme et al.*, *2014*). Active DIYbio projects now range from genetic disease testing to designing water-quality monitoring devices (*Alper*, *2009*; *Jorgensen & Grushkin*, *2011*) and the number and variety of projects carried out by DIYbio groups continues to increase (*Freitag & Pfeffer*, *2013*).

A major goal of this study was to establish collaboration between undergraduate students from University College London (UCL) and members of the public engaged in research at the London BioHackspace Ltd (LBHS). The purpose of the collaboration was to foster skills exchange between UCL and LBHS and also to provide LBHS researchers access to

facilities at UCL with the required legal permissions to perform DNA recombination, an activity which was not possible at the LBHS during the period of the project. To achieve these aims UCL and LBHS collaborated to attempt construction of a shaker-incubator device for cultivation of *Roseobacter* using only publically available components. LBHS members designed a strategy for isolation and subcloning the *anf1* gene from *O. indolifex* into a BioBrick^TM plasmid backbone. UCL students and LBHS members also attempted to transform *R. denitrificans*, *O. indolifex* and *D. shibae* and characterise these strains with respect to cryopreservation and resistance to an antibiotic commonly used to select for retention of recombinant plasmids.

## MATERIALS AND METHODS

Unless otherwise stated, all growth media and solutions were sterilised by filtration or autoclaving. All reagents used to isolate or manipulate DNA were certified as molecular biology grade by the supplier.

### Safety considerations

Standard operating procedures and risk assessments were developed prior to the performance of all procedures. The *Roseobacter* strains investigated require the lowest level of containment, Level 1, as defined by the Advisory Committee on Dangerous Pathogens (ACDP), part of the United Kingdom (UK) Health and Safety Executive (HSE). The template DNA extraction procedure below is a modification of the commonplace method described by *Sambrook & Russell* (*2001*) but revised to omit phenol:chloroform due to the absence of organic chemical storage or manipulation facilities at the London BioHackspace at the time of this work. *The efficiency of this modified procedure for extraction of genomic DNA is doubtful and the procedure is reported here in principle as an illustration of the constraints that must sometimes be negotiated to prioritise safety when working in community laboratories.

All authors of this work were aware of the illegality of unauthorised environmental release of genetic modified organisms in the UK and regarded the terms of their status either as members of the London BioHackspace or as UCL staff or students as a *de facto* formal commitment to ensure no such release was attempted. It is also important to consider that a significant body of research is still required to determine if a genetically modified marine bacterium could establish itself and persist in natural environments. The modes of modification considered in this work would inevitably exert a metabolic burden on host cells, reducing their fitness for natural habitats compared to their wild type competitors.

### Bacterial strains and plasmids

Three *Roseobacter* strains were obtained from NCIMB Ltd (Aberdeen, Scotland): *R. denitrificans* OCh114 (*Shiba*, *1991*), *O. indolifex* HEL-45 (*Wagner-Döbler et al.*, *2004*) and *D. shibae* DFL 12 (*Biebl et al.*, *2005*). The *E. coli* strain W3110 (*Bachmann*, *1972*) was sourced from historic stocks available at UCL.

BioBrick^TM formatted plasmids, pSB3C5 (EU496103) and pSB1C3 (AF532313), were supplied by the Registry of Standard Biological Parts (Massachusetts, USA). Plasmids

pHD1313 (*Alibu et al.*, *2005*), pRPGFPSIR2rp3 (*Borg*, *2015*), pUBeK (*Borg*, *2015*) and pAO815 (Invitrogen, California, USA) were also used.

## Recombinant DNA procedures

Plasmids were purified from cells using standard commercial 'Mini Prep' kits such as the KeyPrep Spin Plasmid DNA Mini Kit Pk100 (Anachem Ltd., Luton, UK). Standard molecular biology techniques were used for restriction digests of plasmids and preparative polymerase chain reactions (PCRs). Preparative PCR was performed at LBHS using a Perkin Elmer (Beaconsfield, UK) Thermal Cycler 480 device. The forward primer used for amplification of *anf1* had the following sequence, with the single-underlined text indicating the Eco RI site, the bold text indicating the Not I site, the double-underlined text indicating the Xba I site and the text in lower case characters indicating the bases complimentary with the *anf1* ORF:

GTTTCTTCGAATTC**GCGGCCGC**TTCTAGAGGCAAGGGAatgcaagacagc. The reverse primer used for amplification of *anf1* had the following sequence, with the single-underlined text indicating the Pst I site, the bold text indicating the Not I site, the double-underlined text indicating the Spe I site and lower case characters again indicating bases complimentary with the *anf1* ORF: GTTTCTTCCTGCA**GCGGCCGC**ACTAGTAGCCTctacttcatcagccgtttg. These sequences include restriction sites in non-coding regions to ensure the amplified *anf1* gene fragment is compatible with the BioBrick™ standard. Linearised pSB1C3 plasmid as template was PCR-amplified using the primer, 'SB-prep-3P-1': gccg**ctgcag**tccggcaaaaaa, which anneals at the pSB1C3 PstI site (in bold) and the primer, 'SB-prep-2Ea': at**gaattc**cagaaatcatccttagcg, which anneals at the pSB1C3 EcoRI site (in bold). The amplified pSB1C3 fragment and *anf1* amplicons were cut with EcoRI and PstI and ligated. All primers were supplied by Eurofins MWG Operon (Ebersberg, Germany).

## Preparation of material containing sufficient template DNA for preparative PCR*

Due to the lack of equipment for safe handling of phenol:chloroform mixtures at LBHS, the following procedure was followed in the hope of deriving sufficient template genomic DNA from *O. indolifex* culture. 1.5 volumes of 100% ethanol and 0.1 volumes of 3 M sodium acetate were added to 400 μL of *O. indolifex* culture followed by mixing with a vortex for 5 s. This material was then placed in a −20 °C freezer overnight then centrifuged at 12,000 RPM for 20 min. The supernatant was decanted and replaced with 1 mL 70% *v/v* ethanol followed by centrifugation at 12,000 RPM for 10 min. The supernatant was decanted and the pellet air-dried and suspended in 30 μL water before use as PCR template. The mass of genome DNA that may have been extracted was not measured. An aliquot of this material was used as template in a PCR reaction with primers specific for the *O. indolifex anf1* gene. Successful amplification of a DNA fragment of expected size indicated that some template *O. indolifex* gDNA template was present. However, this may have been due to the persistence of intact cells that were disrupted subsequently by the 95 °C denaturation step of PCR. Until further work is carried out, we do not currently propose this preparation method to others as an efficient step for gDNA extraction.

## Cell cultivation

Unless otherwise stated all cell cultivation was performed using static and shaking incubators in the UCL Department of Biochemical Engineering, such as the Memmert High Precision Incubator (Memmert GmbH, Schwabach, Germany) and the Kuhner ISF-1-V Climo-Shaker Incubator (Adolf Kuhner AG, Basel, Switzerland) respectively.

## Components used in construction of the 'UCLHack12' shaker-incubator device

The UCLHack12 Public shaker-incubator device (Fig. 5) was constructed at LBHS using the following components: an Arduino Esplora micro computer (Dangi internet Electronics S.L., Almunecar, Spain), a Worldwide Travel Multi-Voltage Power Supply (Maplin Electronics Ltd., Rotherham, UK), an electronic motor (Maplin Electronics Ltd., Rotherham, UK), a 9 V PP9 battery (Maplin Electronics Ltd., Rotherham, UK), two $210 \times 148$ mm cardboard sheets (Ryman Ltd., Cheshire, UK), four $8.4 \times 44.5$ mm springs (Maplin Electronics Ltd., Rotherham, UK), two pencils (Ryman Ltd., Cheshire, UK), two L/C 10/0.1 mm cable wires (Maplin Electronics Ltd., Rotherham, UK), 1 mm diameter copper metal wire (Minsets Ltd., Herts, UK) and a 42L Cool Box (Argos Direct, Stafford, UK) to act as an outer chassis and containment barrier.

## Cryopreservation of cells using glycerol

For all three *Roseobacter* strains a commercial Marine Broth (MB) 2216 was used for cultivation in liquid culture and a commercial MB agar (both Becton–Dickinson, Le Pont de Claix, France) used for growth on plates. Typically, colonies from MB agar plates or 6 μL of growth culture were used to inoculate 6 mL MB in a 50 mL Falcon tube. Inoculants were then incubated for 12–16 h with 200 RPM shaking at 37 °C until typically $OD_{600} = 1$–2 was achieved. After this 1.6 mL sterile 80% *v/v* glycerol was added, and mixed by pipetting up and down. The resultant 17% *v/v* glycerol solution was then divided into 380 μL aliquots and stored separately in labelled tubes at −80 °C. The above procedure was also used for preparation of *E. coli* glycerol stocks using Luria-Bertani (LB) liquid medium and agar plates (Sigma-Aldrich, Munich, Germany).

## Antibiotic sensitivity

For each strain, 100 μL of cells from a glycerol stock were used to inoculate 100 mL of MB (LB for *E. coli*) in a 0.5 L conical flask which was then incubated at 37 °C for 12–16 h with 200 RPM shaking until $OD_{600} \approx 4$. Four 20 mL aliquots were taken and to each was added 20 mL MB containing twice the intended final chloramphenicol concentration indicated in Fig. 3. The 40 mL culture was mixed by brief, gentle swirling and then split into 16 aliquots of 2 mL, each in 15 mL Falcon tubes. All tubes were incubated at 37 °C with shaking at 200 RPM and two 2 mL cultures removed at the indicated time points for $OD_{600}$ measurement followed by disposal. For ampicillin sensitivity 100 μg/mL was used as previous work by *Piekarski et al.* (2009) suggested this as a minimum inhibitory ampicillin concentration for use with *Roseobacter* clade bacteria.

## Preparation of *Roseobacter* and *E. coli* cells competent for plasmid uptake by heat shock

*Roseobacter* strain cells were streaked from a glycerol stock onto a non-selective MB Agar plate and grown for 12–16 h at 37 °C. A single colony picked from the plate was used to inoculate 5 mL of MB in a 50 mL Falcon tube before 12–16 h incubation at 37 °C with 200 RPM shaking until $OD_{600} = 1$–2. 1 mL of this culture was used to inoculate 100 mL MB, in a 0.5 L conical flask, which was further incubated under the same conditions until an $OD_{600}$ of 0.3 was reached. The culture was transferred to two pre-chilled, sterile 50 mL tubes and incubated on ice for 10 min. A five minute 4,000 RPM centrifugation at 4 °C was used to pellet cells. The supernatant was then removed and the pellet resuspended in a 10 mL ice-cold solution of 0.1 M $CaCl_2$ and 15% *v/v* glycerol and incubated on ice for 30 min. This centrifugation step was repeated and the final cell pellet resuspended in 1 mL ice-cold 0.1 M $CaCl_2$/15% *v/v* glycerol solution, then divided into 100 µL aliquots in pre-chilled Eppendorf tubes and stored at –80 °C. The above procedure was also used for the preparation of competent *E. coli* cells, replacing MB with LB in all steps.

## Plasmid transformation by heat shock

Aliquots of cells putatively competent for transformation by heat shock were removed from storage at –80 °C and placed on ice. A maximum volume of 5 µL of plasmid solution was pippetted onto still-frozen competent cells before incubation of 45 min on ice. After this tubes containing now-thawed cells and plasmid were placed in a 42 °C water bath for 10 min to cause heat-shock then transferred to ice for two minutes before addition of 1.3 mL of MB. The solution was transferred to a 15 mL Falcon tube and incubated at 37 °C with 200 RPM shaking for an hour. This material was transferred to a 1.5 mL Eppendorf tube and a 14,000 RPM centrifugation for two minutes used to pellet cells. After removal of the supernatant, the cell pellet was resuspended in 100 µL MB then spread onto selective MB Agar plates. Resultant colonies were assessed after 12–16 h static incubation at 37 °C and again at 24 and 48 h time-points. This procedure was also performed using *E. coli* cells by replacing MB with LB in all steps.

## Preparation of *Roseobacter* cells competent for plasmid uptake by electroporation
### Method A

A modified version of the method reported by *Piekarski et al.* (*2009*) was used. 50 µL of *Roseobacter* strain glycerol stock was used to inoculate 50 mL MB in a conical flask. This inoculum was then incubated at 37 °C with 200 RPM shaking for 12–16 h to an $OD_{600}$ of 1–2. After this a 15 min, 4,000 RPM centrifugation at 4 °C was used to pellet cells. The supernatant was removed and the cell pellet was resuspended in 10 mL pre-chilled 10% *v/v* glycerol. A further four rounds of the same centrifugation, supernatant removal and pellet resuspension prodecure were then performed. The final pellet was resuspended in 1 mL 10% *v/v* glycerol and divided into 50 µL aliquots in pre-chilled 1.5 mL Eppendorf tubes for immediate use.

### Method B

Cells were prepared as in Method A except all cultivation steps were performed at 30 °C and cells were pelleted after having grown to $OD_{578}$ of 0.5.

### Method C

The method reported by *Sambrook & Russell* (*2001*) was used. 10 µL of *Roseobacter* strain glycerol stock was used to inoculate 10 mL MB in a 15 mL Falcon tube. This inoculum was then incubated at 37 °C with 200 RPM shaking for 12–16 h to an $OD_{600}$ of 0.5–1.0. 4 mL of this material was used to inoculate 400 mL MB in a 2 L conical flask which was incubated as above until an $OD_{600}$ of 0.5–0.6 was reached.

The conical flask was then chilled on ice for 30 min and the 400 mL of inoculum transferred to a pre-chilled 0.5 L centrifuge bottle and centrifuged at 4 °C for 15 min at 6,000 RPM. The supernatant was removed by aspiration and 400 mL of ice-cold sterile distilled $H_2O$ was used to resuspend the cell pellet. This centrifugation, supernatant removal and pellet resuspension procedure was repeated twice before the pellet was finally resuspended in 50 mL ice-cold 10% *v/v* glycerol solution. This was then transferred to a pre-chilled 50 mL centrifuge tube and centrifuged at 4 °C for 15 min at 6,000 RPM. The supernatant was removed using a pipette and the pellet resuspended in 2 mL of ice-cold 10% *v/v* glycerol. This was then divided into 50 µL aliquots in pre-chilled 750 µL PCR tubes, which were kept on ice and used immediately. This process was also used for *E. coli* cells by substituting MB with LB.

## Plasmid transformation by electroporation

A maximum volume of 5 µL of plasmid solution was added to 50 µL putatively competent cells in a pre-chilled 0.2 cm pulser cuvette (Bio-Rad, California, USA). The mixture was then pulsed in a Gene Pulse Xcell$^{TM}$ System (Bio-Rad, California, USA) typically using a field strength of 0.5–3.0 kV, capacitance of 25 µF and resistance of 200 Ω. Alternative settings are also discussed in the Results section. After electroporation 1 mL chilled MB was added immediately to the cuvette. For electrocompetent cells generated using Methods A and C, the entire cuvette contents was decanted to a 15 mL Falcon and incubated at 37 °C for 12–16 h with shaking at 250 RPM. For electrocompetent cells generated using Method B, 1 mL 1.7% *w/v* sea salts (S9883 Sigma-Aldrich, Munich, Germany) was added to the cuvette, decanted and split into two 0.5 mL aliquots, each on a 15 mL Falcon. 0.5 mL of water was added to one of these aliquot to give a 0.85% sea salts solution.

For both MB and sea salt solutions, a 100 µL aliquot was spread onto an MB agar plate containing 17 µg/mL chloramphenicol and incubated at 37 °C for 12–16 h with shaking at 250 RPM. Colonies were counted the next day and the presence or absence of plasmid confirmed by mini prep, agarose gel analysis and spectrophotometry.

## RESULTS AND DISCUSSION

### Participants in this work

The practice of synthetic biology is at its most 'open source', we suggest, when performed by people who are not molecular life science 'experts', such as graduate or postgraduate

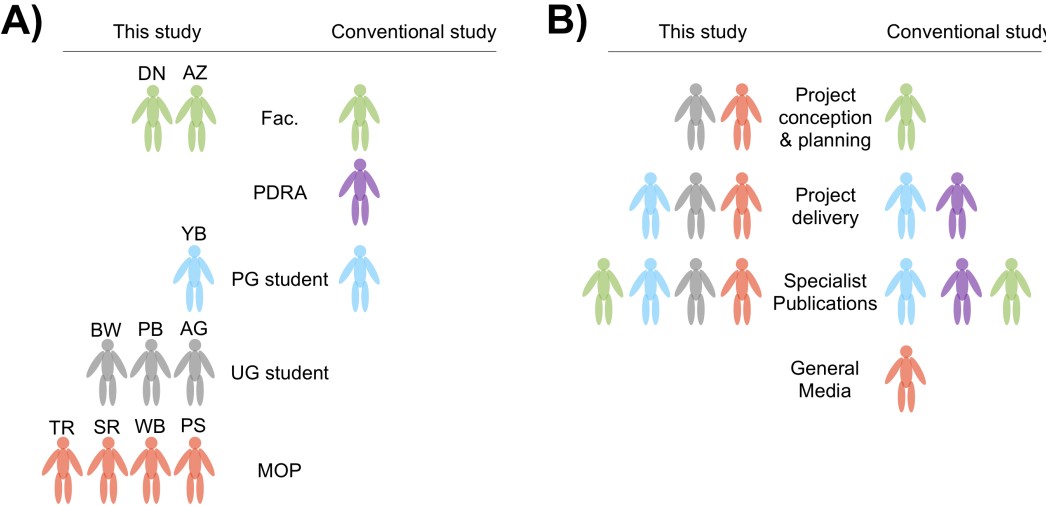

**Figure 1 Project conception and organisation.** Diagrammatic summary of the conception and delivery of this study with respect to the roles of the people involved. Colours signify the following—green, member of university faculty; purple, post-doctoral research assistant (PDRA); blue, post-graduate (PG) research student; grey, undergraduate (UG) student and orange, member of the public (MOP). Acronyms indicated the following authors—YB, Yanika Borg; AG, Aurelija Marija Grigonyte; PB, Philipp Boeing; BW, Bethan Wolfenden; PS, Patrick Smith; WB, William Beaufoy; SR, Simon Rose; TR, Tonderai Ratisai; AZ, Alexei Zaikin and DN, Darren N. Nesbeth. Note that YB, SR, TR, WB and PS do not have a life science first degree. (A) Typically, a conventional study (right hand side of the panel) is performed by members of university faculty as principal investigators (PIs), PDRAs, and PG research students, while UG students or MOPs tend not to be involved in primary research roles. This study (left hand side of the panel) featured no PDRAs, three UG students and four MOPs. (B) A conventional study (right hand side of the panel) is conceived and planned by PIs who bid for funds to support a given number of PDRAs and PG research students to carry out the work. Resultant data is then written up by the research team in a manuscript that is submitted to specialist scientific journals. In this conventional model MOPs will only learn of the research via general media such as national newspapers. This study (left hand side of the panel) was conceived and planned solely by UG students and MOPs. A PG research student then assisted with experimentation and UCL faculty members assisted with writing up the resultant data.

students of the field, or life science professionals in academe or industry. Toward this end it is advantageous that this project was planned and conceived by four members of the public and three undergraduate (UG) students, as summarised in Fig. 1. We define 'member of the public' (MOP) in this instance as somebody who does not have a life science degree and is not studying for a life science degree. Of the three UG students, two were studying life science degrees and one a degree in computer science. The MOPs and UG students met during the summer of 2012. The impetus behind their meeting was the 2012 International Genetically Engineered Machines competition.

The UG students received introductory training from postgraduate students at University College London in standard molecular biology technques sufficient to perform BioBrick$^{TM}$ assembly. The UG students then shared their knowledge with MOPs who had previously been trained using Internet sources and fellow LBHS members from various backgrounds. Together the UG students and MOPs defined this project and led the design of the experiments. One doctoral student assisted closely in this study, Yanika Borg (YB), whose first degree is in mathematics and statistics and who had less than one

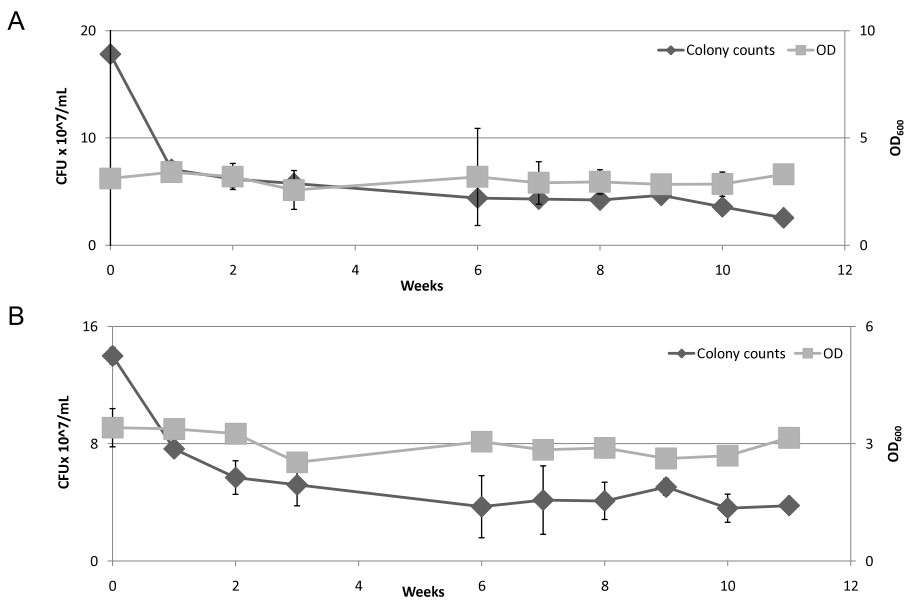

**Figure 2** **Robustness of *Roseobacter* to glycerol-based cryopreservation.** Aliquots of *R. denitrificans* (graph A) and *O. indolifex* (graph B) cells were stored in 17% *v/v* glycerol solution at −80 °C. At the indicated time points post-storage, aliquots were thawed, diluted and either spread on MB agar plates or used to inoculate liquid MB medium. 24 h later CFU counts (diamonds) and $OD_{600}$ values (squares) were plotted. Error bars indicate standard deviation over two biological repeats.

year of molecular biology experience during this work. YB proviced assistance on the work reported in Figs. 2–4, with purely UG students and MOPs involved in the work reported in Figs. 5 and 6. With respect to experimentation, UCL faculty member, Darren N. Nesbeth provided only safety supervision and logistical support—but did not conceive the study, choose the topic or design the experiments. Darren N. Nesbeth and fellow UCL faculty member Alexey Zaikin also provided advice on the drafting of the manuscript.

## Robustness of *Roseobacter* strains to glycerol cryopreservation

At the outset of this study we were aware that a number of organisations maintain commercial culture collections, such as NCIMB Ltd (Aberdeen, Scotland) and professional research laboratories, and routinely cryopreserve *Roseobacter* and *E. coli* strains. We first aimed to establish whether standard procedures could also achieve reliable preservation of *Roseobacter* strains *O. indolifex* and *R. denitrificans* when performed by students and members of the public, who are relatively inexperienced with respect to microbiological techniques, and in a community laboratory setting. Citizen scientists at LBHS typically perform experiments at evenings and weekends on 1–2 occasions per week to fit around employment or other interests. When time and equipment are scarce, separate incubators, or incubator rotas, enabling a choice of cultivation at 25 °C, 30 °C or 37 °C are somewhat of a rarity. The majority of experiments conducted at LBHS are with *E. coli* cultivated at 37 °C. Because of this we attempted to cultivate *Roseobacter* cells at 37 °C, as successful growth at this temperature would afford maximum flexibility to researchers investigating

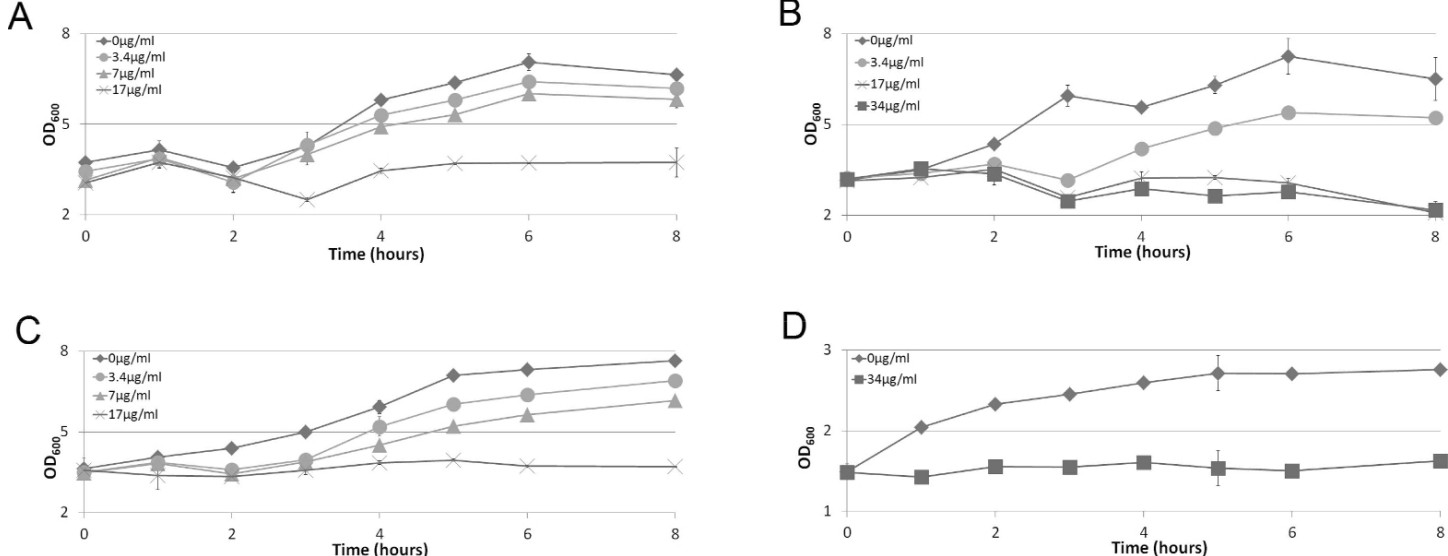

**Figure 3   Chloramphenicol sensitivity of Roseobacter and _E. coli._** Growing cultures of _D. shibae_ (graph A), _R. denitrificans_ (graph B), _O. indolifex_ (graph C) and _E. coli_ (graph D) were cultured in the presence of no antibiotic (diamonds), or chloramphenicol at 3.4 µg/mL (circles), 7 µg/mL (triangles), 17 µg/mL (crosses) or 34 µg/mL (squares). $OD_{600}$ was then measured at the time-points indicated. Error bars indicate standard deviation over two biological repeats.

_Roseobacter_ at LBHS. All four _Roseobacter_ strains in this study grew successfully at 37 °C in MB (Figs. 2 and 3).

We measured the ability of _O. indolifex_ and _R. denitrificans_ cultures to survive cryopreservation by measuring how many colony forming units (CFU)/mL remained within glycerol stock solutions each week over 11 weeks (Fig. 2, diamond data points). Initial 6 mL cultures of _O. indolifex_ and _R. denitrificans_ were cultivated as detailed in 'Materials and Methods'. Duplicate 50 µl aliquots were removed and diluted 10,000 fold in MB, using a 1,000-fold dilution of 50 µl cell suspension into 50 mL followed by a 10-fold dilution of 5 mL aliquot of this dilution into 45 mL medium in 50 mL Falcon tubes. For both strains, 10 µL of the 10,000-fold diluted material was spread onto duplicate non-selective MB agar plates. The plate was incubated for 12–16 h at 37 °C after which the number of colonies, typically 100–200, was counted and used to calculate the number of CFU in the original undiluted culture sample. These data were plotted at week zero in Fig. 2. The remainder of the 6 mL culture was split into separate Eppendorf tubes for cryopreservation in glycerol as described in the 'Materials and Methods'. A single Eppendorf tube containing the glycerol stock was removed and thawed on ice each week for 11 weeks. CFU/ mL was measured as above, taking into account dilution due to glycerol addition.

We also quantified the ability of _O. indolifex_ and _R. denitrificans_ cultures to be revived from cryopreservation by measuring how much growth was achieved by glycerol stocks used as inoculant each week over 11 weeks (Fig. 2, square data points). Duplicate 50 µL aliquots were removed from an initial 6 mL culture and each used to inoculate 10 mL MB which was then incubated at 37 °C for 12–16 h with shaking at 200 RPM and the $OD_{600}$ measured. These data were plotted at week zero in Fig. 2. 50 µL aliquots were

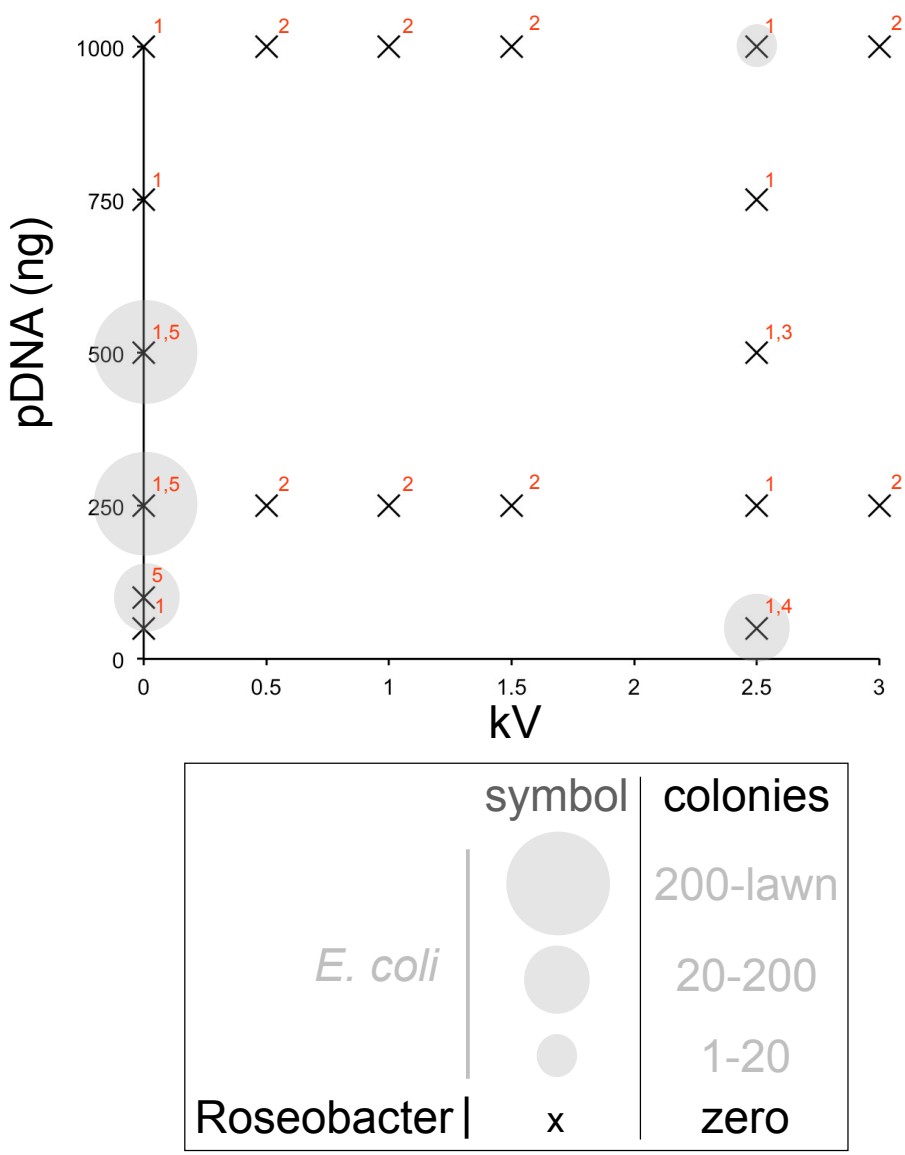

**Figure 4 Plasmid transformation method performance map.** Bubble chart in which transformation results are plotted as crosses in the absence of Roseobacter strain transformant colonies and closed grey circles when *E. coli* colonies are observed (circle size indicating colony numbers, see key), as a function of plasmid mass (*y* axis) and kilovolts used (*x* axis, zero for heat shock method). *E. coli* are only used where indicated by grey circles. Red numbering indicates the plasmids, method and Roseobacter strains used in each experiment: (1) Method A was used to attempt transformation of *D. shibae*, *R. denitrificans* and *O. indolifex* with pSB3C5 at the indicated voltage and zero volts as control. (2) Method A was again used in an attempt to transform *D. shibae* with pSB3C5. (3) Method C was used to attempt transformation of *D. shibae* with plasmids numbered 2–5 in Table 1, using the appropriately selective MB agar plates. (4) Method B was used to attempt transformation of *D. shibae* with plasmids numbered 2–5 in Table 1. (5) The heat shock method was attempted for transformation of *D. shibae* with pSB3C5.
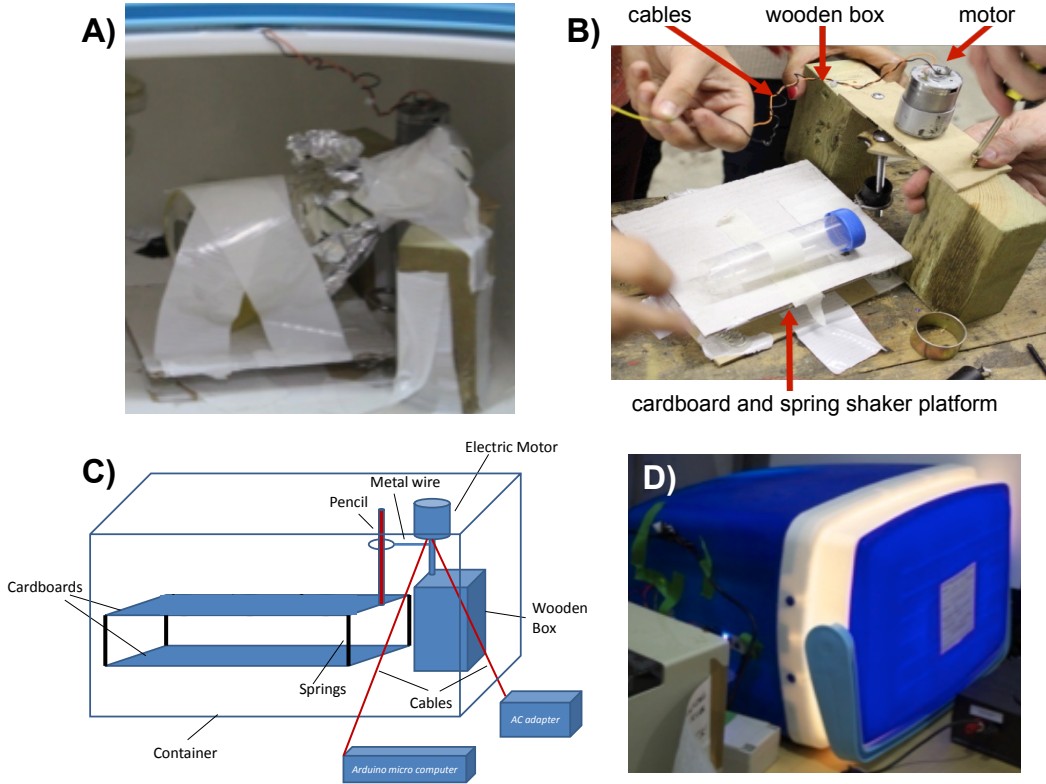

**Figure 5** **The UCLHack12 open source incubator-shaker device.** (A) Photograph of the device to which a 250 mL conical flask is attached with tape, *in situ* within the 42 L Argos Cool Box. (B) Photograph showing the wooden blocks used as a supportive housing for the Maplin Electronics electric motor, the electric cables and the spring-assisted shaker platform to which a 50 mL Falcon tube is attached with tape. (C) Schematic diagram of the arrangement of components used to construct the device. (D) Photograph of the closed 42 L Argos Cool Box containing the functioning device.

subsequently removed from a glycerol stock each week and used to inoculate 10 mL MB prior to incubation and $OD_{600}$ measurement as above.

In Fig. 2 the survival of cells is plotted as CFU/ mL as a function of weeks spent at –80 °C. Both *R. denitrificans* (Fig. 2A) and *O. indolifex* (Fig. 2B) show a sharp initial decrease in cell survival after one week in a glycerol stock solution at –80 °C, compared to their starting viability before glycerol addition (week zero). For both species, from weeks 1–11 there is a shallow downward trend from $\approx 8 \times 10^7$ CFU/ mL to $\approx 4 \times 10^7$ CFU/mL. The ability of the cells to be revived by growth in liquid culture is plotted in Fig. 2 as $OD_{600}$ after 12–16 h growth as a function of weeks at –80 °C. Unlike survival performance, which decreases over time, revival remains effectively constant throughout the 11 weeks for both species.

These data indicate that in our hands the *Roseobacter* strains remain viable over 11 weeks and are likely to remain viable over much longer periods, particularly when considering revival by growth in liquid culture. Interestingly, *R. denitrificans* and *O. indolifex* both grew well at 37 °C. This observation confirmed reports by *Bruhn et al.* (*2006*) and *Christie-Oleza et al.* (*2012*) that *Roseobacter* strains can be cultivated over a broad range of temperatures. By contrast, *Lafay et al.* (*1995*) reported that incubation at 37 °C failed to elicit growth
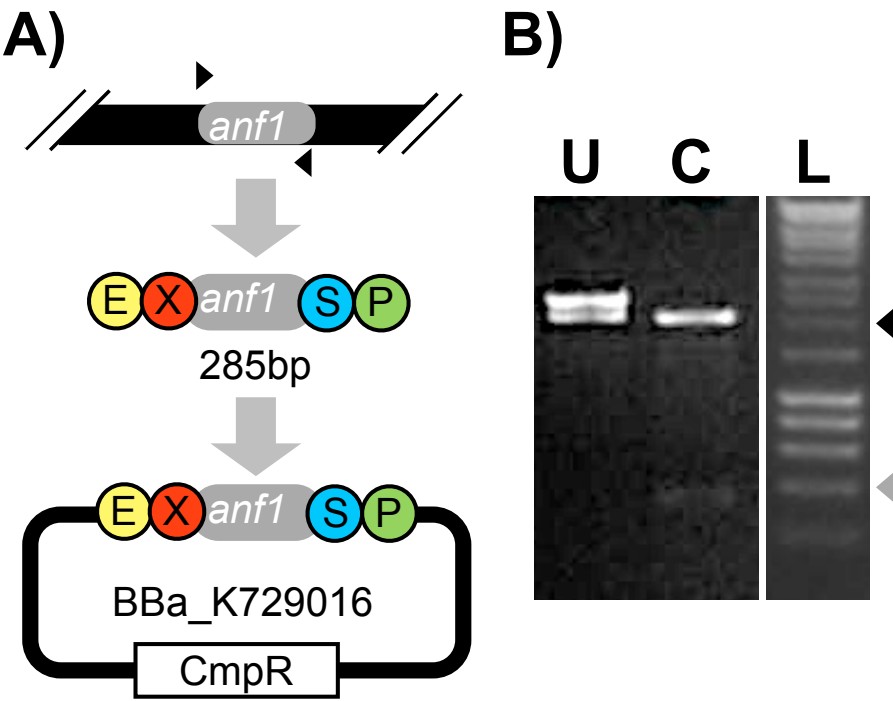

**Figure 6** **Amplification and subcloning of *anf1*.** (A) A schematic overview depicting the primers (black triangles) used to PCR-amplify the *anf1* ORF to successfully yield an amplicon fragment encoding EcoRI and XbaI sites upstream and SpeI and PstI sites downstream of the *anf1* ORF. The fragment was then subcloned into pSB3C1 to form the BioBrick™ plasmid, BBa_K729016. (B) Consists of photographs of agarose gel lanes in which uncut BBa_K729016 (lane U), BBa_K729016 digested with EcoRI and PstI enzymes (lane C) and DNA ladder (lane L) have been run. The 400 bp and 2,000 bp bands of the ladder are indicated by grey and black triangles respectively.

of *R. denitrificans*. This suggests *Roseobacter* strains may be sensitive to unidentified variations in handling and provenance to a degree that is not typically observed with biotechnology 'workhorse' organisms such as *E. coli*. Preliminary data available here http://2012.igem.org/Team:University_College_London/Research/MarineBacteria, show growth of *O. indolifex* and *E. coli* at 37 °C and 25 °C in 10 mL culture volumes in 50 mL Falcon tubes. At both temperatures, *O. indolifex* shows more growth in MB 2216 than LB and *E. coli* shows more growth in LB than MB 2216.

## Chloramphenicol sensitivity of *Roseobacter* strains

Chloramphenicol is used to select for retention of the widely used pSB3C5 BioBrick™ plasmid backbone that encodes a chloramphenicol resistance gene as its selectable marker. In our view a major step in establishing an organism as a tractable chassis for synthetic biologists use is to demonstrate compatibility with standard tools that have been widely adopted by the synthetic biologist community. To determine if pSB3C5 BioBrick™ could be maintained by the *Roseobacter* strains in this study we sought first to establish the minimum chloramphenicol concentration required to suppress cell growth, being mindful of the 30 µg/mL minimal inhibitory chloramphenicol concentration reported by *Piekarski et al.* (*2009*). For *E. coli* and all three *Roseobacter* strains, growing

cultures were diluted and varying concentrations of chloramphenicol added before growth was monitored by $OD_{600}$ measurements.

Figure 3 shows measurements of cell growth over an eight hour period after chloramphenicol addition. The data indicates that at least 17 µg/mL of chloramphenicol is required to suppress growth of all *Roseobacter* strains, with lower concentrations permitting growth. Where a higher chloramphenicol concentration is used, as in the case of 34 µg/mL chloramphenicol used for *R. denitrificans*, no greater effect is observed in comparison to 17 µg/mL (Fig. 3B).

## Attempted plasmid transformation of *Roseobacter* strain cells

We mapped a number of different parameters with respect to plasmid transformation of *Roseobacter* strain cells in an attempt to establish a robust and straightforward protocol for engineering these cells (Fig. 4). *Piekarski et al.* (*2009*) methods were attempted alongside a selection of alternative approaches. Figure 4 is a bubble chart that provides an overview of the conditions for which we established transformation efficiency for *E. coli*, *Roseobacter* strains or both. Although no transformation of *Roseobacter* strains was observed we suggest the selection of conditions investigated can serve as a starting point for future efforts to identify a suitably robust and effective protocol. Below we discuss the individual experimental conditions of note.

Electroporation using method A (see 'Materials and Methods') was used to attempt to transform all *E. coli* and all *Roseobacter* strains with 50 ng, 250 ng, 500 ng and 1,000 ng of pSB3C5 with voltages of zero (as control) and 2.5 kV (Fig. 4). For *D. shibae* further voltages of 0.5 kV, 1 kV, 1.5 kV and 3 kV were used with 250 ng and 1 µg of pSB3C5 which features the p15 ori. *Piekarski et al.* (*2009*) highlighted *ori* type as a potentially important factor for plasmid propagation in *Roseobacter* species. As such electroporation Method B was used for transformation of *D. shibae* with pSB3C5 and plasmids 2–5 as numbered in Table 1, which possess a range of selectable markers and origins of replication (*ori*). 50 ng of each plasmid was used with a voltage of 2.5 kV (Fig. 4). Method C was also used for transformation of *D. shibae* at 2.5 kV with 500 ng each of plasmids 2–5 (Table 1).

Transformation was also attempted using heat shock with *D. shibae* and 100 ng, 250 ng and 500 ng of pSB3C5 (Fig. 4). Control transformations (Fig. 4, https: //figshare.com/s/3fd20f74ef890472198e) were performed that omitted some or all of the following: antibiotic selection, plasmid or heat shock. All such control experiments yielded either colonies or clear plates along with expectations and indicated that none of the methods or plasmids is inherently cytotoxic.

In addition to *Piekarski et al.* (*2009*), other groups have reported successful transformation of *Roseobacter* genera such as *Ruegeria mobilis* (*D'Alvise & Gram*, *2013*) and *Silicibacter* (*Miller & Belas*, *2006*). These studies involved cell cultivation using Heart Infusion Broth and yeast extract, both of which differ significantly from the marine growth media used for *Roseobacter* strains.

## Cultivation of *O. indolifex* using the UCLHack12 incubator-shaker device

In our hands *Roseobacter* strains were in effect resistant to transformation. This led us to reflect that development of such protocols may be addressed better in future with high throughput automation approaches, using equipment such as the Tecan Genesis RMP device (Tecan, Männedorf, Switzerland) which was recently kindly donated to LBHS by Paratopes Laboratories (London, UK).

Although confident that simple and reliable protocols for *Roseobacter* transformation will be identified in the near future using automated, high throughput experimentation, we were also curious to investigate whether characteristic *Roseobacter* phenotypes, such as cold tolerance, could be ported to *E. coli*. As a first step toward this goal a subgroup of the authors of this paper, consisting only of LBHS members, designed a cloning strategy to isolate the antifreeze *anf1* open reading frame from *O. indolifex*. To isolate this gene, *O. indolifex* was successfully cultivated at LBHS using an open source device for growth of *O. indolifex* cells.

The device, designated the 'UCLHack12', was constructed using the components indicated in the 'Materials and Methods' assembled in the configuration illustrated in Fig. 5. A battery-powered Arduino microcomputer was programmed to effect rotary movement of the electric motor. The electric motor was attached to a metal ring with an inserted pencil such that circular motion of the motor caused the pencil to move back and forth. The pencil was firmly connected to a cardboard panel that as a result was moved back and forth by the motion of the pencil (Figs. 5B and 5C). The cardboard panel attached to the pencil was also attached to a second, lower cardboard panel by four metal springs to amplify lateral rocking motion. Adhesive tape was used to secure a 250 mL conical flask (Fig. 5A) or 50 mL Falcon tubes (Fig. 5B) to the upper cardboard panel. Closing the lid of the container box provided sufficient insulation to maintain an internal temperature of 28–30 °C. The UCLHack12 device was designed and constructed at LBHS solely through collaboration of undergraduate students and citizen scientists.

At LBHS, 5 mL of MB in a 50 mL Falcon tube was inoculated using an *O. indolifex* glycerol stock and incubated for 12–16 h in the UCLHack12 device at 30 °C with agitation of approximately 100–150 RPM. Incubation in this manner achieved an observable increase in medium turbidity, although regrettably optical density measurements were not taken during or after incubation. One repeat of this procedure was performed before unknown LBHS users then dismantled the UCLHack12 device without the knowledge of the authors.

## Amplification and subcloning of the *O. indolifex anf1* gene

400 µL of the *O. indolifex* culture grown in the UCLHack12 device was conditioned using the ethanol-based procedure to liberate genomic DNA and render it accessible to oligonucleotide binding as part of preparative PCR. A subgroup of the authors of this study led by LBHS members used web-based software tools to locate the *anf1* gene within the *O. indolifex* genome (ENA, http://www.ebi.ac.uk/ena/data/view/EDQ05862) and designed primers to achieve both amplification of the *anf1* open reading frame (ORF) and subsequent sub-cloning into pSB1C3, via the restriction sites required for compatibility

with the BioBrick$^{TM}$ standard. The primers were delivered to UCL and transported to LBHS for use in preparative PCR along with the *O. indolifex* genomic DNA template material. At LBHS agarose gel electrophoresis revealed that a 285 bp DNA fragment, the expected size for the *anf1* gene, had been successfully amplified by PCR. A small mass of linearised pSB1C3 backbone was also successfully amplified by PCR. A ligation reaction was performed at LBHS to combine the *anf1* gene fragment into pSB1C3 using standard molecular biology techniques. If successful this reaction would yield a BioBrick$^{TM}$ plasmid encoding the *anf1* open reading frame (ORF) as a BioBrick$^{TM}$ 'part' with the code BBa_K729016 according to the notation of the Registry of Standard Biological Parts. The ligation reaction was transported to UCL and used to transform competent *E. coli* W3110 strain cells. This step was necessary as the LBHS was not at the time licensed to cultivate organisms harbouring modified genetic material. Transformation at UCL using a standard heat shock method was successful: plasmid DNA was isolated from transformants and positively identified as BBa_K729016 by restriction digest (Fig. 6). BBa_K729016 was subsequently the first BioBrick$^{TM}$ part submitted to the Registry of Standard Biological Parts by a community laboratory led and run by members of the public.

## CONCLUSIONS

### *Roseobacter* cultivation and storage

Working as a team of students and 'DIY' biologists, we established that *Roseobacter* strains *O. indolifex* and *R. denitrificans* are robust to glycerol-based cryopreservation at −80 °C over 11 weeks. We also successfully cultivated *O. indolifex* in the UCLHack12 incubator-shaker device that we assembled entirely from publicly available components. Items such as the UCLHack12 can help address issues of resource limitation that are common for newly founded community laboratories. These achievements can now inform future efforts to establish *Roseobacter* strains as synthetic biology chassis and also foster investigation of other strains that are relatively under-explored due to their unknown or challenging genetic tractability.

### Establishing recombinant DNA techniques in Roseobacter

We determined a minimum chloramphenicol concentration required to arrest growth of *O. indolifex* and *R. denitrificans* to inform future protocols for plasmid propagation in these strains. We also mapped the performance of a broad set of electroporation conditions using a battery of plasmids with a range of replication origins and selectable markers, none of which yielded transformants. We are confident that further work using equipment such as a Tecan Genesis RMP device will establish *Roseobacter* strain transformation protocols that are both effective and sufficiently robust to be usable by researchers in a diverse range of settings with respect to available training and facilities.

### De-skilling, bio-geoengineering and governance

Earth's biosphere has a profound impact on the planet's surface geology and meteorology through processes such as the sulphur cycle (*Alcolombri et al.*, *2015*). In theory, synthetic biology could be used to modify the biosphere via the re-writing of bacterial and eukaryotic

genomes. In practice, as genome editing and writing technologies are ported to a wider range of chassis, many of which will be robust to marine or extreme environments, so the dilemmas and choices raised by synthetic biology are set to increase. Bio-geoengineering and 'de-skilled' molecular biology remain future possibilities rather than current realities. Despite this it is still tempting to envisage uncomfortable scenarios in which so-called hobbyists, benevolent or otherwise, tinker with technologies, such as genome editing or the environmental release of genetically modified organisms, whose dangers they do not fully understand (*Jansen et al.*, *2014*). Part of our intention with this study was to present an example of responsible research conduct in a community laboratory that was both safe and aspired to novelty and impact.

Upon submission of BBa_K729016 to the Registry of Biological Parts we took steps to raise awareness of the first 'Public BioBrick™' through various media channels as part of the activities of the 2012 UCL 'Plastic Republic' iGEM team. Events included an exhibition at the Grant Museum, UCL, with the biological art practitioner, Dr. Howard Boland (*Boland*, *2013*, doctoral thesis). We also sought to capture the experiences of researchers involved in this work, using techniques reported by *Tweddle et al.* (*2012*), to inform future projects. We invited LBHS members to complete a written survey (*Jordan et al.*, *2011*) and video interviews to assess whether participation in the project had impacted their research skills and outlook on collaboration with academia. A preliminary record of these data can be found at the Wiki homepage of the 2012 UCL iGEM team (http://2012.igem.org/Team:University_College_London/HumanPractice/DIYbio/Evaluation). A clear outcome of the survey was that citizen scientists felt both their bench and study design skills had increased. The need to work to deadlines was perceived as onerous. UCL students' responses to the survey indicated they felt that community laboratories such as LBHS had broadened their perception of the settings in which scientific research can take place within society.

One obvious conclusion from this work is that genetic modification of *Roseobacter* cannot yet be regarded as a de-skilled procedure (*Tucker & Danzig*, *2012*). However, this study also serves to illustrate that wider participation in science and engineering is now a reality (Fig. 1). Familiar modes of debate regarding science and technology separate industrial and academic researchers from 'members of the public'. This view is becoming obsolete as ever more powerful research tools, such as cloud-based biological experimentation, genetic modification and high throughput automation, become affordable and accessible to the 'general public'. Opportunity brings risk. The possibility of unwanted outcomes caused by environmental release of designed organisms, borne around the globe by winds and oceans, must of course be a central concern for all researchers regardless of the setting in which their research is performed.

## ACKNOWLEDGEMENTS

Dr. Paola R. Gomez-Pereira (National Oceanography Centre, Southampton, United Kingdom) and Miss Yeping Lu contributed to developing the concept of this study.

### Funding

Funding from the Wellcome Trust, EPSRC, BBSRC and the UCL Faculty of Engineering is gratefully acknowledged. The funders had no role in study design, data collection and analysis, decision to publish, or preparation of the manuscript.

### Grant Disclosures

The following grant information was disclosed by the authors:
Wellcome Trust.
EPSRC.
BBSRC.
UCL Faculty of Engineering.

### Competing Interests

The authors declare there are no competing interests. Patrick Smith, Simon Rose, Tonderai Ratisai and William Beaufoy are employees of London BioHackspace, London Biological Laboratories.

### Author Contributions

- Yanika Borg and Aurelija Marija Grigonyte conceived and designed the experiments, performed the experiments, analyzed the data, wrote the paper, prepared figures and/or tables, reviewed drafts of the paper.
- Philipp Boeing, Bethan Wolfenden, Patrick Smith, William Beaufoy, Simon Rose and Tonderai Ratisai conceived and designed the experiments, performed the experiments, analyzed the data, contributed reagents/materials/analysis tools, reviewed drafts of the paper.
- Alexey Zaikin contributed reagents/materials/analysis tools, reviewed drafts of the paper.
- Darren N. Nesbeth analyzed the data, contributed reagents/materials/analysis tools, wrote the paper, prepared figures and/or tables, reviewed drafts of the paper.

### Data Availability

Raw data plotted in Figs. 2, 3 and 4 are provided at Figshare: https://figshare.com/s/7a7db66d6e2cb28c047a; https://figshare.com/s/97c68054418faad1f277; https://figshare.com/s/3fd20f74ef890472198e. The research in this article did not generate other raw data.

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
