# Peer review of "Open source approaches to establishing Roseobacter clade bacteria as synthetic biology chassis for biogeoengineering"

_PeerJ, doi:10.7717/peerj.2031_

## Round 0.1 · original submission · Major Revisions

The reviewers bring up a number of significant questions/concerns, both with regard to experimental methods and validity of results. In addition, questions as to the broader context of this work have also been raised. Please address these concerns in your revisions.

·

Basic reporting

The article addresses an important and poorly studied problem in brininging new researchers into productive work with new organisms. A paragraph or two pointing to important literature for novice workers would help -- many of the references are likely already present. A discussion of safety issues would be helpful, even if to reassure readers that there were few issues.

Experimental design

Line 194: This appears to be a very unusual method of preparing genomic DNA. I’m not even sure it should work. There appears to be little that will lyse the cells — the method seems more like a method of precipitating existing DNA in solution. Are you sure a step is not missing? With PCR conditions, of course, a PCR might work anyway, with lysis of the cells happening in the cycler at 96.
Line 217: It would be good to be more specific about the exact culture medium used. Is there a BD number?
Line 347: It is surprising that you found growth at 37 when it is not reported elsewhere. I would be concerned about culture contamination. Does colony morphology match with other examples? Does E. coli grow on your MB plate? A 16S PCR and sequence would verify this but I understand you are working with limited resources.
Line 377: Why did you choose pSB3C5 as your plasmid to transform? Is p15 ori the correct one? Is there evidence it will propagate?
Line 384: With transformation with these other plasmids, was culturing done with ampicillin plates? There is not mention of this, and the plasmids don’t carry chloramphenicol resistance genes, to my knowledge. Would this explain why no transformants were present?
Line 391: Another possible explanation for colonies would be integration of the cat gene into the chromosome, so no plasmid was found. Alternatively, a failure of mini prep. What made you conclude that the colonies were false positive?
Line 442: details are missing here, and are likely important. The primers listed on line 190-191 to amplify the pSB3C5 plasmid have the order of restriction sites appropriate for amplifying the insert, but not the vector. If used to amplify the vector, then cut with an enzyme to ligate the anf1 gene fragment, the result will be an incorrect set of restriction sites in the final vector. This won’t matter until further assemblies with that part are attempted. This issue will likely be caught by resequencing at IGEM HQ in the spring.
Line 725: I would identify the source and literature reference for each of the other plasmids in the table.
Figure 4: not clear what the agar plates are doing in the 4C. They don’t need to be shaken ??
Fig 5: A) I doubt the restriction order in the bottom section is correct (likely an extra XbaI site ahead of EcoRI site, and an extra SpeI site after the PstI site). But I hope I’m wrong. The short band in lane C of B) is almost invisible in my print out. (I think it’s there).

Validity of the findings

The cryro results look solid. It would be interesting to quantify what happens with multiple freeze/thaw cycles and longer periods.
The transformation results are disappointing, but it is a start, and someone has to start. I would be trying to exactly replicate the previous successful work at this point, and then go from there.
Growing strains is an important first step, and they managed to do this. It would help to experiment more with viable temperature ranges, salt concentrations and media components.
A little bit of sequencing would go a long way here, both to verify the constructs they have made, and to verify the purity and identity of the strains they were working with.

Additional comments

Line 116: order of restriction sites is incorrect. Should be EcoRI; NotI; XbaI and SpeI; NotI; PstI. I would reference Shetty et al. Methods in Enzymology paper.
Line 129: A ref to the cold gene would help.
Line 185 - 191: underlining the restriction sites in sequence, and lowercasing the primer binding region will help readers. On line 191 there appears to be an extra lower case t int he middle of the sequence.
Line 247: from not form
I’d prefer to see spaces between numbers and units such as 10 ml not 10ml; 10 minutes not 10minutes.
Line 741: Should be 3.4 ug/mL (missing /) also in following places.
Line 748: kilovolts (no space)

·

Basic reporting

The paper seeks to demonstrate the applicability of open source approaches to synthetic biology and bioengineering.
This is not the classical aim of a scientific paper. It raises many questions: Why should science be “democratized” in the first place? Which aspect of science can safely be “democratized” and still yield scientifically sound results? How is reproducibility achieved? What is the rational of using the equipment described in the paper, and not something perhaps more functional and still “open source”. Is “open source” a quality in itself, or should it have a serious reason?
Part of the “open source” concept was to have work done by “citizen scientists”. This, again, raises a number of questions. Has this concept any advantage for the reported scientific goal? If not, then what was the goal and might it have been achieved by other means as well? How were “citizen scientists” recruited, how were they trained, how was the quality of their work assessed, how was their level of understanding assessed? Did they just perform the work like technicians? Who exactly were the “team of students” and “DIY scientists”? How was the success of “democratizing science” finally evaluated?
If the “open source” aspect is an important part of this publication, then it should be described just as extensively as the purely technical part and should be reviewed. However, none of these points are described or discussed.

Experimental design

When we exclude the aspect of “democratizing science” and focus exclusively on the “classical” science in the paper, then it comes down to the following:
1. Cryopreservation works as expected for three Roseobacter species.
2. Genetic transformation of Roseobacters cannot yet be regarded as de-skilled procedure (as expected as well).
3. Cloning a “bio-brick” from a Roseobacter was possible.
From the point of view of classical science none of this is a true scientific advancement. All three experiments would be ideal for a laboratory course with students (or pupils), once they have been familiarized with basic techniques and with working under sterile conditions. Cloning a gene into an E coli plasmid is a a standard procedure. The transformation procedure published by Piekarski et al. 2009 could not be reproduced. Novel data for the scientific community to continue with have not been generated. Thus the paper does not report a scientific achievement and would not be publishable according to current practice.

Validity of the findings

A closer look at the data shows that important technical aspects of the work are questionable.
1. How was it assured that the right culture grew and not a contaminant? At one point, an OD600 value of 4 is reported, which is very unusual for Roseobacters (line 228).
2. Growing the bacteria in a closed falcon tube might be causing a problem because they are aerobic. How much culture volume was in the falcon tube? A 2 ml culture might be okay in a 50 ml falcon for 24 h.
3. The amplified band in Fig. 5 has approximately the right length. It could still be the wrong product. It would be necessary to sequence it to prove that it is indeed the anf1 gene.

Additional comments

1. Abstract line 25 and elsewhere: “Roseobacter clade bacteria comprise 15-20% of oceanic bacterio-plankton communities”. This statement is wrong. They comprise 15-20% in some regions of the ocean at certain times of the season in certain depths, and 1% or less at other times of the year, other regions of the ocean, and other depths. The correct statement is “up to”. It may seem a minor detail but in fact it is not (thinking about controlled release, ecosystem engineering etc. it is important to get the basic facts right).
2. Line 171. All three citations are wrong. R. denitrificans was first described by Shiba 1991, O. indolifex by Wagner-Döbler et al. 2004, and D. shibae by Biebl et al. 2005.
3. Line 193 ff. Preparation of material containing template DNA for preparative PCR: The described methodology is very unusual. If it has been developed in the course of this project, then it should be validated and compared to established methods.
4. Line 220. Why were these bacteria, which are marine strains and not human commensals like E. oli, grown at 37°C?
5. Line 296. Which sea salt solutions are meant?
6. Line 393. “Roseobacter” refers to a large number of genera, including Ruegeria and Silicibacter.
7. Line 401. What exactly could be improved with the TECAN device? High throughput automated cultivation will achieve nothing if there is an un-identified biological problem.
8. Line 427. “Incubation in this manner achieved an observable increase in medium turbidity.” This is a very vague statement to demonstrate the quality of the device. Which OD was reached? Was it reproducibly reached? Did it only work for O. indolifex, or for all three strains? At which point during the 12 – 16 h was it reached? Was the turbidity perhaps caused by medium components? “Normally” one would record OD values over a period of 24 h in the inoculated flask compared to the control and repeat the experiment two or three times. And compare it to growth in a traditional, not “open source” device. If it is enough to vaguely see turbidity, then that means that “democratic science” is neither precise nor reliable nor reproducible.
9. General comment: The framework of the story is about cleaning the ocean from plastic waste using engineered bacteria that have been constructed using “democratic science”. There is a reason why the release of recombinant bacteria into the environment is currently under very stringent control. To me, the open source approach is really frightening in this context. However, it might not be too dangerous because the likelihood of it being effective is extremely small. Cloning a gene from a Roseobacter into E. coli as the first step of constructing a bug to clean the ocean to me sounds like “Vow I know how to bike. How about going to the moon?” But, who knows.

---

## Round 0.2 · accepted · Accept

Thank you for taking the time to address the concerns expressed by the reviewers.